# A Miniaturized Full-Ground Dual-Band MIMO Spiral Button Wearable Antenna for 5G and Sub-6 GHz Communications

**DOI:** 10.3390/s23041997

**Published:** 2023-02-10

**Authors:** Tale Saeidi, Ahmed Jamal Abdullah Al-Gburi, Saeid Karamzadeh

**Affiliations:** 1Electrical and Electronics Engineering Department, Faculty of Engineering and Natural Sciences, İstinye University, Istanbul 34396, Turkey; 2Center for Telecommunication Research & Innovation (CeTRI), Fakulti Kejuruteraan Elektronik dan Kejuruteraan Komputer (FKEKK), Universiti Teknikal Malaysia Melaka (UTeM), Durian Tungal, Malacca 76100, Malaysia; 3Millimeter Wave Technologies, Intelligent Wireless System, Silicon Austria Labs, 4040 Linz, Austria; 4Electrical and Electronics Engineering Department, Faculty of Engineering and Natural Sciences, Bahçeşehir University, Istanbul 34349, Turkey

**Keywords:** button spiral antenna, leaky-wave antenna, 5G, sub-6 GHz, WBAN, wearables

## Abstract

A detachable miniaturized three-element spirals radiator button antenna integrated with a compact leaky-wave wearable antenna forming a dual-band three-port antenna is proposed. The leaky-wave antenna is fabricated on a denim (ε_r_ = 1.6, tan δ = 0.006) textile substrate with dimensions of 0.37 λ_0_ × 0.25 λ_0_ × 0.01 λ_0_ mm^3^ and a detachable rigid button of 20 mm diameter (on a PTFE substrate ε_r_ = 2.01, tan δ = 0.001). It augments users’ comfort, making it one of the smallest to date in the literature. The designed antenna, with 3.25 to 3.65 GHz and 5.4 to 5.85 GHz operational bands, covers the wireless local area network (WLAN) frequency (5.1–5.5 GHz), the fifth-generation (5G) communication band. Low mutual coupling between the ports and the button antenna elements ensures high diversity performance. The performance of the specific absorption rate (SAR) and the envelope correlation coefficient (ECC) are also examined. The simulation and measurement findings agree well. Low SAR, <−0.05 of LCC, more than 9.5 dBi diversity gain, dual polarization, and strong isolation between every two ports all point to the proposed antenna being an ideal option for use as a MIMO antenna for communications.

## 1. Introduction

Due to the wireless communications industry’s rapid growth, the need for high data rates, huge channel capacity, and high dependability has resulted in a notable increase in the use of multiantenna systems. In a mobile device with limited space, it becomes more challenging to pack several antennas tightly together while maintaining a high isolation level [1]. Using multiple antennas to boost channel capacity without using more frequency or transmit power is known as multiple-input-multiple-output (MIMO) technology. By utilizing spatial and polarization diversity with several antennas at the transmitter and receiver, capacity gain can be achieved. Polarization diversity is preferable in real applications because the antennas can be collocated [2], which overcomes the constraint of the spacing of the antenna elements. The use of MIMO antennas in numerous applications, including wearables and wireless body area networks (WBANs), was made possible by their promising qualities, including diversity characteristics (pattern, polarization, and gain diversity) and the properties above. 

There is a lot of promise for WBANs in the fields of identification systems, personal healthcare, and sports training. Flexible materials are favored for user comfort reasons. According to earlier research, diversity is crucial in WBANs because the constantly moving human body may shield or obstruct the signal [3]. Space diversity, pattern diversity, and polarization diversity are the three different types. In other circumstances, it is more challenging to incorporate spatial diversity because of how little room there is in the human body. Therefore, a low profile is necessary for wearable systems. For example, a patch antenna or a horizontal dipole/monopole antenna with a reflector may be employed [4]. However, it is far more challenging to implement vertical polarization, which is particularly advantageous, for example, in wearable MIMO systems [5].

Modern wireless communication systems also frequently need to incorporate several standards with different operating frequencies and protocols. Antennas with a range of bands and capabilities are required for this. Multiband and dual-polarized antennas with perpendicular polarizations are also gaining popularity in WBAN systems because they can reduce polarization loss brought on by the motions of the human body. The multipath fading issue can also be resolved, and the data rate increased via polarization diversification, which is where MIMO antennas come in. Wireless MIMO communication methods improve multipath propagation performance. A MIMO system’s data rate is inversely proportional to the number of antennas it employs.

Additionally, the data rate and channel capacity in MIMO systems will rise proportionally as the number of antennas increases. The number of antennas will enhance the spatial correlation between the two received signals/antennas and reduce the distance between them. The error rate performance lowers as the spatial correlation rises. Therefore, low spatial correlation and strong isolation between the antennas are the main requirements for MIMO systems to improve the error rate performance of wireless communication systems. However, achieving the necessary isolation levels between antennas with coupled ground structures might be difficult. The isolation between the antennas in MIMO systems is improved by using a defective ground structure (DGS) with an antenna. Other approaches have also been put forth, including split ring resonators (SRR) and electronic band gaps (EBG). There have been several different MIMO antenna configurations proposed, including a two-element MIMO resonating at a single band with a connected ground plane, a three-element MIMO antenna with pattern and polarization diversity [2,6,7,8,9,10], a dual band [11,12,13], and a wide band/UWB [14,15,16,17].

Wearable antennas play an increasingly significant role in WBAN applications. These antennas must be lightweight, flexible, and withstand the harsh dynamic operating environment and deformations brought on by on-body circumstances such as bending or crumpling. In addition to mechanical flexibility, radiation agility quality is desirable for many wearable systems [18]. Conducted textile antennas [19,20], flexible substrate-based antennas [21,22], button antennas [23], and miniaturized integrated antennas have all been studied in the past. A wider frequency tuning range would be advantageous in terms of frequency agility to account for the variance in on-body operating conditions and surroundings and cover various communication protocols. Most of the aforementioned on-body antennas are also constructed on textile substrates. The textiles of the body would be folded and twisted in this case, which would change the resonance frequency, bandwidth, and radiation efficiency of textile antennas [24,25,26,27]. Instead of employing textile substrates to design on-body antennas, this issue is resolved using clothing elements such as buttons. The solid button antenna is easy to attach to clothing with traditional tailoring and is resistant to crumpling and bending. The SAR is also reduced when button antennas are distanced from human tissues. 

Wearable button antennas with several bands have been researched. Polarization variety has not been, unfortunately, attained therein. The button antenna is also frequently stiff, which guarantees better radiation performance than a wearable antenna made entirely of textiles. Due to their ability to provide detachable RF connections with adequate RF performance, they have also been proposed as a practical and cost-effective radiofrequency (RF) connection solution for wearable devices. However, wearable antennas made of metal have also been developed for textile buttons.

Additionally, for strong performance, button antennas can rarely deform and are significantly less susceptible to any change in temperature or humidity [28,29,30]. Research was also conducted on a button antenna that was circularly polarized to lessen polarization loss. The proposed antenna, however, could only operate in one band, which restricted its uses, and no pattern variety was achieved. Therefore, a single port and just two resonances were used to feed a modular textile antenna design developed for wearable electronics. A wearable button antenna sensor for dual-mode wireless information and power transmission and a wideband button antenna was also designed. For 5 GHz WBAN applications, it was likewise based on distinctive feature mode theory and an omnidirectional circularly polarized button antenna. Nevertheless, they showed some BW flaws, such as poor endurance and lack of circular polarization (CP) [31,32,33,34]. 

As stated previously, for a wearable antenna to perform optimally for WBAN applications such as personal healthcare and sports training, it must be able to provide high data rate communication, large channel capacity, pattern diversity, polarization diversity, gain diversity, miniaturization, multiband, and dual polarization. In addition, it should also offer multipath fading resolving, low spatial correlation, and strong isolation between the antennas to improve the error rate performance, an example of which is a decrease in transmission errors.

Therefore, designing a button antenna that can be considered part of a cloth, easily attached and detached, and not be defected by bending can be integrated with a textile antenna to resolve the challenges above. It also should have the following specifications for WBAN applications: (1) multiband to support multiple communications benchmarks, (2) dual-polarized pattern diversity to boost connection speeds and eliminate polarization discrepancy, and (3) miniaturized button size to maximize accessibility, which is quite innovative and rational. According to the authors, these three characteristics have not yet been attained concurrently for a wearable button antenna. Additionally, the problem could become immensely complicated when (1) both on and off-body channels must be taken into account, (2) potential muscle activity, and (3) various poses.

Integration of a button antenna and a textile MIMO antenna having circular polarization or polarization diversity (dual or triple polarization) can solve these problems. As mentioned before, when a direct path between the receiver and the transmitter is present, it is possible to enhance the signal-to-noise ratio in the transmission medium and attain a better power balance between the vertical and horizontal polarization. Moreover, this can be achieved using circular polarization (CP) transmission rather than linear polarization (LP) transmission. This advantage can be used even more to increase diversity gain. Wearable circularly polarized antennas have already been the subject of several efforts. However, the off-body channel, which corresponds to the transmission between wearable devices and an indoor base station, is the primary priority of most communications [18,35,36,37]. 

A miniaturized circularly polarized multiple-input multiple-output (MIMO) antenna with 40 mm^2^ × 40 mm^2^ was designed for wearable biotelemetric devices. However, they did not use flexible textile materials and offered lower gain [38]. Another sub-6 GHz ISM-band flexible wearable MIMO antenna array for wireless body area networks (WBANs) and biomedical telemetry devices are designed on a rigid substrate. They could obtain high gain using electromagnetic bandgap (EBG) and metasurface (MTS). Nevertheless, the dimensions were not miniaturized [39]. A four ports MIMO antenna was designed on textile for 5G communication to improve the isolation. However, the dimensions were increased to do so [40]. In addition, a two-port flexible MIMO antenna with the defected ground was designed for the ISM band to improve the isolation [41]. Several MIMO antennas were designed to resolve some of the wearable MIMO antennas’ challenges mentioned above. For example, a partially ground was used to improve the isolation for a wearable UWB MIMO antenna [42], a novel low-profile 5G MIMO antenna was designed on a rigid substrate to exhibit dual-band frequencies for 5G NR-n2 band (1.9 GHz) and safety band (ITS-5.9 GHz) in vehicular communication [43], and a quad-port smart textile UWB MIMO antenna with dimensions of 50 mm^2^ designed for biohealthcare applications [44] (A comparison of these recent works are given in Table 1). 

An innovative dual-band button antenna that supports 5G and sub-6 GHz communications and is a viable contender for deployment in WBAN applications is described in this research. Incorporating a spiral three-element detachable button antenna and a leaky-wave antenna with a full ground to fulfill the SAR criteria led to dual-band functioning. The antenna sensor generates a broadside radiation pattern with circular and dual polarization. The proposed antenna’s monopole-like resonant mode, which produces an omnidirectional radiation pattern in the lower operating zone, further sets it apart from earlier crossed dipole antennas [51]. This radiation pattern resembles a monopole and is used in the World Interoperability for Microwave Access (WiMAX) spectrum for on-body data transmission (at 3.45–3.85 GHz). Broadside radiation with dual and circular polarization is used for transmission and receiving in the WLAN frequency range of 5.4–5.85 GHz. The proposed antenna’s specific absorption rate (SAR) level in both bands is lower than the cut-off points for the European standard. 

The system analysis and design configuration of the unique wearable button antenna is examined for WBAN applications in Section 2, after a brief and educational introduction in Section 1. Next, Section 3 determines a complete evaluation of the antenna and the findings of the simulation and measurement. Finally, Section 4 offers the final observations and conclusions.

## 2. Antenna Configuration 

The proposed prototype of the dual-band, dual-polarized button antenna, as well as the radiation modes for on-body and off-body communications, are presented in this section. The proposed wearable MIMO antenna is a three-port leaky-wave MIMO antenna integrated with a three-element spiral button antenna to operate for 5G and sub-6 GHz communications. At first, the button antenna and the MIMO LWA are designed separately at the desired frequency bands and then integrated. The button antenna contains three spiral elements (to obtain CP and multiband features) at the front and three CSRR elements at the back (to improve the impedance bandwidth level, radiation efficiency, gain, and surface current since the spirals are connected to them by pins). The button antenna operates at three resonances, and it is detachable. This is to make the whole antenna system (integrating both button and LWA) radiate towards the broadside. The MIMO leaky-wave antenna is also designed to resonate in the same bands as the button antenna. The leaky-wave antenna is chosen since it can steer and scan the beam with frequency. Transversal and longitudinal slots load the leaky-wave MIMO to improve the leaky-wave matching and solve the OSB issues that, intrinsically, every leaky-wave antennas have. Integrating the button antenna and the MIMO LWA is achieved in a way that offers constructive radiation characteristics toward the broadside. The suggested antenna’s mechanism is clarified by examining the current and electric-field distributions. Optimization on an inhomogeneous phantom (chest, arm, and shoulder) is also studied to simulate the genuine on-body application contexts and have more realistic circumstances. 

### 2.1. Antenna Geometry 

Figure 1b,c shows the perspective view of the proposed button antenna integrated with the leaky-wave antenna (LWA). It can be worn on clothes, as shown in Figure 1c. The button antenna consists of three spiral elements at the front and three complementary split ring resonators (SRRs) at the back. Figure 1e,f shows the spiral button antenna with the circular SRR structures connected to the spirals with the shorting pins (brown dots in Figure 1f). Afterward, the three spirals are connected to the wearable textile LWA with three nails (black dots in Figure 1e). The leaky-wave antenna indicated in Figure 1a,d is fed through a transmission line. It goes through one rectangular resonator loading with transversal and longitudinal slots between the transmission line and these two rectangular patches (Figure 1d). This integration focuses on the gain and the radiation on the button antenna (mainly towards the broadside) for the desired frequency bands and WBAN application.

#### 2.1.1. Spiral Button Antenna 

Wideband, bidirectional radiation, and circular polarization technologies have extensively used planar spiral antennas because of their construction, corresponding to frequency independence [52]. Although a planar spiral antenna emits radiation in two directions, wearable antennas and WBAN applications require an antenna with unidirectional radiation qualities. There are several varieties of spiral antennas, including the Archimedean spiral, logarithmic spiral, square spiral, star spiral, and more. Archimedean and logarithmic spirals are relatively common among the most studied frequency-independent antennas. There are several methods for turning a planar spiral antenna’s bidirectional radiation into unidirectional radiation, including backing the antenna with an absorbing cavity, using reflective artificial electromagnetic material as a reflector, and backing the spiral with a perfect electric conductor (PEC) cavity [53].

Consequently, we used a full ground for the LWA antenna. The substrate’s bottom layer loaded the button spiral with comparable SRR rings and shorting pins. Additionally, a sort of UWB antenna is the spiral antenna [54,55]. The equiangular spiral antenna is widely recognized for its large bandwidth and circular polarization properties. However, if a wave absorber is utilized, it may lose half of the EM energy due to the bidirectional radiation pattern [56]. The separation between the equiangular spiral antenna and the bottom is fixed if the reflector is only a flat surface. The ground modifies the equiangular spiral antenna’s far-field pattern. Thus, it is unidirectional rather than bidirectional, and the BW is limited. 

The spiral button antenna is designed on a circular polytetrafluoroethylene (PTFE) substrate with a permittivity of 2.1, a loss tangent of 0.001, and a thickness of 0.508 mm. The metal SRR rings are printed at the bottom layer of the substrate. 

The most common production processes for the radiator components of spiral antennas are printing and wiring [57]. Because polychlorinated biphenyls (PCBs) are so quickly produced and have incredible accuracy, we used printing in this investigation. On a spherical Rogers 5880 substrate, the antenna’s radiator consists of three periodic Archimedean spirals (Figure 1). The design of the Archimedean spiral radiator in open space is first taken into account, and they rely on the equations as follows [58]: (1)$$r=r_{in}+\alpha \theta$$
(2)ractive=λf2π
(3)$$r_{in}<r_{10\;GHz\;active\;region\;}$$
(4)$$r_{out}>r_{2\;GHz\;active\;region\;}$$
(5)r22=rout>λflow2π
where *f* is the operating frequency, r_active_ region is the radius of the active region at the operating frequency f (3.2 GHz), and r_in_ and r_out_ the inner and outer radii of the spiral arm, respectively; *r* is the radius of the spiral arm for the winding angle *θ* (rad); *α* (mm/rad) is its growth rate; the spiral antenna’s outer diameter (r_2_), lowest working frequency wavelength (λ_f_ Low), spiral arm width (w_4_), gap (g_7_). Apart from the abovementioned equations, more equations exist to calculate the spiral sections [59]. For instance, the proportionality constant (r_0_) is determined from the width of each arm, w_7_, and the spacing between each turn, g_7_, which for a self-complementary spiral is given by
(6)r0=g7+w1π 

The strip width of each arm can be found in the following equation.
(7)g7=rout−rin2N
where N is the number of turns.

Each spiral follows a periodic order regarding the number of turns and the angle. Hence, they can be more useful when the pattern and steering angle (α) are required and can be optimized until the desired variables are achieved. The impacts of all these design factors on the antenna’s impedance bandwidth, impedance matching, AR, and radiation efficiency were calculated using parametric sweeps in the 3D computer simulation technology (CST) Microwave Studio environment. These parameters included the growth rate and the inner radius r_in_. Only a brief description of the impact of these factors is provided in this work, which does not offer a parametric investigation. For instance, a spiral antenna radiator must consider the growth rate, which determines how quickly the spiral antenna flares or expands as it rotates. A lower spiral turn count from r_in_ to r_out_ indicates a greater growth rate. Low spiral turn estimates result in inadequate radiation at the active area, which prevents any radiated signal from reaching the active regions of undesirable modes. The accumulation of the critical mode and undesirable mode pattern causes the symmetry to be broken [58]. It should be noted that a spiral antenna on a substrate with a relative dielectric constant larger than 1 has a lower impedance than an antenna in free space when choosing the growth rate. To produce an impedance more significant than the one in free space, it needs to be considered. The AR values should also be taken into consideration. R_in_, another crucial parameter, determines the top operating frequency limit. Thus, it was also believed that r_in_ would undergo a parametric assessment. As r_in_ increases, the antenna’s real and imaginary impedances rise, especially at high frequencies, and the AR results decrease. The diameter of the feeding zone is about the same as that of the active region at the highest frequency limit [58], which is the primary cause of the inferior AR findings. The degradation is visible with a r_in_ parameter of 2.7 mm greater than 1.2 mm because the diameter of the feeding region at the 1.2 mm value of the r_in_ parameter is still distant from the diameter of the active region’s 10 GHz ring.

Consequently, the smallest r_in_ should be picked. The coaxial cable used in the Phelan balun’s output ports has a diameter of 0.86 mm. Therefore, 0.86 mm is the minimum distance between the output cables’ center conductors. A value of 0.4 mm was chosen for r_in_.

It should be mentioned that the largest spiral is fed first to obtain the lowest resonance, around 3.4 GHz. Then, the other two spiral growth factors are added to obtain the different resonances and improve the antenna’s circular and dual polarization characteristics. Figure 2b illustrates the spiral button antenna connected to a piece of textile and fed through an SMA port. The S-parameter results and the extracted refractive index (n), permittivity ($\varepsilon_{r}$), and permeability ($\mu_{r}$) (real and imaginary parts) of the modified CSRR unit cell are presented in Figure 2c. With their analogous circuit models, which show the SRRs’ whole concept, the CSRR’s operating principles may be properly understood. Electromotive power (EMF) is observed at the SRR when an external field is applied along the Z-axis. This EMF connects two metallic rings concentrically spaced apart and is currently generated in them. The current moves from one ring to the next through the capacitance formed between the rings’ internal gaps. A parallel LC resonator serves as the SRR’s equivalent circuit. The presence of full inductance is shown by the current flowing from one ring to another (LT). The SRR structure, comprising the two halves and the rings, is the basis for the provided capacitances. Additionally, it includes the capacitances connected to the gaps in the split rings. [60] contains all of the equations for the total capacitance C_.T._ and inductance LT. For the more extensive and two smaller SRRs, the resonant frequencies (f_r_) are 3.4 GHz and 5.6 GHz, respectively [61]. A computer-simulated microwave studio was used to model the SRR operation for additional verification (CST MWS). The SRR construction was first placed close to the transmission line. The two smaller SRRs resonate around 5.55–5.65 GHz, whereas the bigger SRR exhibits a resonance at 3.35–3.45 GHz. The transmission coefficient level shows great levels of near zero, like −2.5 dB (Figure 2d). In addition, the refractive index is much closer to zero at these two desired frequencies. Each unit cell is designed for a specific resonance. The three SRRs were added based on each particular frequency and with an order. They also act like a photonic band gap, which helps suppress unwanted resonances and create the desired resonances. Moreover, they also stop the surface wave and unwanted fringing fields for the button antenna [62]. 

For a better understanding of how the current flows through each spiral and how their capacitive gap can affect that, the surface current distribution of the spiral button without any loading is presented in Figure 3a,b. It depicts that the current flows through the largest spiral and then parasitically through the other two spirals. After a parametric study of the spiral antenna without having SRR [63,64] and pins, a modified circular complementary SRR (CSRR) unit cell consisting of three rings is added at the back based on the surface current (Figure 3c,d) and the electric field around spirals. The three SRR rings exist with periodic growth factors and three radiuses of r_5_, r_6_, and r_7_. Each ring acts like a capacitor parallel to a series resistor and inductor. Each ring is coupled with the other one with a coupling capacitor.

The negative index of the metamaterial (MTM) structure, such as the permittivity, permeability, and refractive index, indicates how effectively it improves the button antenna and LWA’s qualities at these frequencies. Each ring is linked to one using a shorting pin to lessen surface waves and unwanted suppressed fringe fields surrounding the spirals. These pins’ positions were selected based on the Figure 3 surface current distribution around the spirals and CSRR. The BW, gain, radiation, and polarization characteristics of the spiral button antenna are examined, and the critical parameters are optimized to meet the needs. Figure 4 shows the reflection coefficient results of the button antenna with each spiral separately and without the spirals. It is also depicted that when all three spirals exist, all three resonances associated with each split ring operate. One unwanted resonance also occurred, which was removed after integration with the MIMO LWA. The antenna is then integrated with an LWA to increase the directive gain and the beam scanning range and obtain more broadside radiation outward the body. Table 2 indicates the antenna parameter values.

#### 2.1.2. Flexible Wearable LWA 

Before starting the design process of the flexible wearable LWA, it would be informative to briefly investigate the challenges of designing the LWA and how they are faced and resolved in this article. So, the challenges are first described, and then the solution used in this paper is explained. 

A 1-D leaky-wave antenna (LWA) directs the wave in a single direction along the guiding structure. The antenna is typically fed from one end, and the wave propagates via the slots and along the design’s axis [65]. A leaky-wave antenna (LWA) is a traveling-wave device based on a transmission line packed with radiating components [66]. Initially, a rectangular waveguide with slots was used to make them [67]. These slot configurations can be used to estimate the polarization and radiation direction of the LWA [68,69]. 

The proposed LWA includes a rectangular patch fed by a transmission line. The antenna is stimulated from one end and ends with a matched 50-ohm load on the other. Continuous beam scanning of the LWA can also be accomplished using balanced composite right/left-handed (CRLH) transmission lines [70]. A left-handed transmission line can be created by stacking unit cells together, such as a T-shaped electrical circuit made of series capacitors, shunted inductors, and transversal/longitudinal slots. The parasitic influences of the capacitors and the inductor culminate in a combination behavior that gives rise to the so-called artificial composite right/left-handed transmission lines (CRLH) [71,72], making it impossible to implement a fully left-handed transmission line in reality. Backward waves can travel on a pure left-handed transmission line (LHTL). It has a smaller than zero equivalent permittivity and permeability. As a result, the comparable refraction index is also smaller than zero, confirming the object’s left-handedness. Radiation could backfire with this perfect structure [73,74].

The conventional and initial dimensions of the rectangular patch and the feed line of the proposed LWA can be achieved utilizing the basic equations as follows (since these equations are certain principles, only a few of these formulas are given here, and the rest can be obtained from [75,76]):(8)Wp=λ2 (εr+12)−12
(9)$$\varepsilon_{eff}=\frac{\varepsilon_{r}+1}{2}+\frac{\varepsilon_{r}-1}{2}\;\left(1/\sqrt{1+12\left(\frac{h}{w}\right)}\right)$$
(10)$$L_{s}=6h+L,\;\;W_{s}=6h+L$$

After obtaining the conventional LWA’s dimensions, other factors necessary for a leaky-wave antenna are considered. The beam scanning range is among the most crucial components of LWA design. Because of the open-stopband (OSB) phenomenon, the conventional LWA has already produced radiation toward end-fire and backfire but has not yet been broadside [77]. No electromagnetic (EM) power is transmitted into the LWA for broadside radiation since the reflected waves from each unit cell are in phase at the frequency of broadside radiation. By using asymmetrical components, impedance matching, and reflection removal, very few different LWAs have already been presented to lower the OSB [78]. To operate above the cut-off frequency, forward-radiating LWAs with transverse slots, as illustrated in [79], have been designed. 

On the patch fed by the transmission line, longitudinal and transverse slots were combined to overcome this issue. These LWAs frequently emit linearly polarized EM waves with their polarization directions parallel to the longitudinal axes of the antenna and the transmission line. The purpose of this is to conduct reflection canceling so the OSB can be controlled [80]. As a result, the transverse slot pair needs to be changed to permit the radiation of alternate polarization (for example, during the LWA experiment, while Port 1 is active, a dummy with an impedance of 50 ohms is substituted there) [81,82]. The longitudinal slots are designed for varied lengths and widths to enhance the OBS. The transverse slots serve as a sequence of left-handed capacitances (C_L_).

Additionally, it reduces surface waves generated by LWAs to improve isolation between the array’s two antennas. The interference of waves partially transmitted through the partially reflected surface determines the power emitted through the slots. Along with applying these slots, the LWA is used entirely to minimize the OSB and apply more excellent reflection to the broadside.

These slots change the direction of the current and the fields around the patches. The combination of these slots changes the principles of a conventional left-handed transmission line (LHTL) to improve the radiation pattern towards the broadside, not end-fire or backfire. It should be mentioned again that the LHTL can be achieved by stacking unit cells made of a T-shaped electrical circuit made of series capacitors and shunted inductors since a pure LHTL can support backward waves. The propagation constant of the LHTL is 𝛽𝐿 (𝜔) = −1/𝜔$\sqrt{L_{L}C_{L}}$, where 𝐿_𝐿_ and 𝐶_𝐿_ are the inductance and capacitance per unit length, respectively. It has a smaller than zero equivalent permittivity, permeability, and refraction index, confirming the object’s left-handedness. The LHTL configuration combines right- and left-handed transmission lines (CRLH-TLs). The propagation constant is, therefore 𝛽𝐶 = 𝜔$\sqrt{L_{L}C_{L}}$−1/𝜔$\sqrt{L_{L}C_{L}}$, where 𝐿_𝑅_ and 𝐶_𝑅_ are the series inductance and shunt capacitance per unit length, respectively. A CRLHTL is dominant LH mode while 𝑓 < 𝑓_0_, RH mode while 𝑓 > 𝑓_0_, and with infinite wavelength while 𝑓 = 𝑓_0_. 𝛽𝐶 varies from negative to positive with the increase in frequency. This characteristic creates a leaky-wave antenna that can radiate in the backfire and end-fire directions. These justifications apply to conventional systems when broadside radiation is unimportant, and a long scanning beam range is not required. 

Apart from applying these slots, the LWA utilized a full ground to apply more reflection towards the broadside and suppress the OSB. Moreover, it reduces the SAR values, which will be explained later. Several articles utilized SIW techniques and transitions for LWA to oppress the OBS issue. However, it is not utilized in this work since applying the SIW transition is not a good candidate when flexibility and robustness are vital, especially in wearable technologies. As will be seen subsequently, this alteration to the antenna array enhanced its impedance bandwidth and radiation qualities, i.e., gain and efficiency, without enlarging it physically. When the LWAs meet the required BW and resonances, along with the other expectations such as higher gain and higher directive broadside radiation rather than end-fire and backfire, they are integrated with the third antenna to reduce the symmetry of the antenna and form the MIMO antenna. 

A modified rectangular patch antenna fed utilizing a ground coplanar waveguide technique is integrated with the LWAs (in X-axis) antennas from different orientations (Y-axis) to create another polarization. Furthermore, it improves the OSB issue since it is asymmetrical to the antenna structure. To better understand how the LWAs and grounded coplanar waveguide (GCPW) antenna function, the surface current distribution of the LWAs is presented in Figure 5. It shows how the wave propagation starts from Port 1 and travels through the slots to get to the other port or load. Figure 6a–c indicates the simulated prototype of the LWAs and GCPW antenna, respectively. Figure 6d demonstrates the proposed antenna’s reflection coefficient results for each MIMO stage before integration. It presents that the LWA antennas obtained resonance around 3.4 GHz. Nevertheless, the more significant shift happened in the lower band, and the higher resonance was shifted to the higher band around 7 GHz because the spacing and gaps between the feeding lines were not uniform. On the other hand, the GCPW antenna achieved resonance at 3.4 GHz and another at a higher band around 5.5–5.6 GHz. 

## 3. Results and Discussion

The results and discussion section consists of two sub-sections showing the antenna performance in both on- and off-body conditions. The assessment in free space (off-body) is examined first. Then, the antenna is in proximity to the human body to see if it performs well when worn. Figure 7 displays the initial reflection coefficient data for each stage of antenna design, such as before and after integration. 

It depicts the reflection coefficient results of the button antenna with and without SRR rings, the button antenna with SRR and pins, and the MIMO antenna with and without integration of the button antenna. The spiral button antenna with three spirals offered three resonances. After adding the CSRR to the button antenna, resonances are shifted to the lower bands because of the inductive loading. After adding the pins, the passband around 2.5 GHz is suppressed and becomes a dual band. The MIMO and the button antenna are integrated consequently. The locations where the button antenna’s pins are attached to the MIMO are chosen based on the surface current of the MIMO so as not to shift the working bands from the desired bands, as shown in Figure 7. Therefore, the integration of the MIMO and the button antenna offers better performance than the other stages in Figure 7. Because the working bands are still achieved after integration, this integration demonstrates better radiation performance, and beam scanning is presented later within the text. In addition, after integrating the button antenna with the MIMO antenna, both desired frequencies are achieved at 3.4 GHz and 5.4–5.6 GHz.

### 3.1. Antenna Performance for Off-Body Condition

Figure 1 and Figure 8 show the working model created to test the suggested concept. The antenna’s prototype was fabricated using a textile substrate and SheildIt conductive textile. These layers were glued using special glue for pasting plastics on textile fibers. The button antenna was manufactured in the lab using chemicals such as ferric chloride. Then, an HP 8510 vector network analyzer (VNA) with two terminals was used to measure the S-parameter results of the antenna (the antenna is calibrated before starting the measurement). First, Ports 1 and 2 and then Port 3 were connected, and the measurement was performed and extracted for each step. Figure 9 compares the simulation results with the S-parameters measured in the air. The simulated findings and the measured results agree rather well. However, there is a slight disparity in the upper band because it was relocated. It remains in the band, though, and it functions in the appropriate band and application. Every resonance is easily visible for all ports to accommodate the applications’ needed band at 3.4 GHz and 5.6 GHz. The measured overlapping bands for Ports 1 and 2 are 3.30–3.5 GHz and 5.34–5.50 GHz, respectively. These ranges span the 5.3–5.7 GHz WLAN band and the upcoming 5G communication frequency for the internet of things (IoT) [83,84,85].

After the S-parameter results were measured and compared with the simulation results, the antenna’s radiation pattern was also measured using VNA, a P.C. computer, and an anechoic chamber at the working bandwidth of 0.1–10 GHz. The measurement was performed by connecting an antenna to the VNA’s first terminal and another to its second terminal. The S-parameters of an antenna—its reflection and transmission coefficients—were measured once the calibration and frequency band were assigned. Then, the antenna’s radiation pattern was also measured (Figure 10). The reference horn antenna, which connects to the power meter, was fixed on the revolving rod to determine the radiation pattern. The vertical walls of the chamber shown in Figure 10 (the wall on the left) have wheels, which were moved to the right when the rotating rod was adjusted to measure the radiation pattern on the right side. 

In contrast, the proposed antenna was fastened to the rod’s center. Next, the rotating rod was rotated by a motion controller that turns it 3 degrees at a time, stops at each stage, records the information, and then rotates the revolving rod another 3 degrees. The radiation pattern was then generated using the computer’s power meter captured in MATLAB software. Last but not least, the antenna was shifted to the elevation plane, and the measuring procedure was restarted.

The antenna simulated and measured radiation patterns (E and H fields) are depicted in Figure 11. A little and negligible discrepancy occurred between the simulated and measured results (the slight discrepancy in the results might occur due to the fabrication tolerances that might happen during the fabrication process.). Therefore, the proposed antenna can be an excellent candidate for wearable antenna systems and communication since the radiation pattern is all outward of the body and has the main lobe towards almost 0°. 

### 3.2. Antenna Performance for On-Body Condition

The proposed antenna’s performance for the on-body condition in terms of the S-parameters, radiation pattern, gain, efficiency, axial ratio (AR), and SAR value is presented in this section. The simulation setup for the on-body (body tissue and voxel arm, voxel chest, and shoulder) condition is shown in Figure 12. The simulation setup explained in Figure 12a,b on the tissue body consists of four layers of bone (h_1_), muscle (h_2_), fat (h_3_), and skin (h_4_) with dimensions of a × b. The space between the antenna and the body in Figure 12 is less than 5 mm. The tissue thickness values of h_1_, h_2_, h_3_, and h_4_ are 21 mm, 7 mm, 5 mm, and 2 mm, respectively [84,86]. In addition, a body voxel (arm, chest, shoulder) is added to investigate the antenna characteristics more realistically. The voxel body and the proposed wearable antenna are shown in Figure 12c–e. The radial distance of the antenna and voxel is less than 4 mm.

The simulated S-parameter results of the antenna for on-body conditions on both human tissue and voxel body are indicated in Figure 13. Both desired bands function and are in the working BW. However, the bands are shifted more on the voxel than the tissue due to the body conditions and layers. Furthermore, more band notches occurred for the results of both tissue and voxel as compared to the free space outcomes. Nevertheless, the desired bands work according to the application’s requirements. 

Figure 14 and Figure 15 show the antenna’s radiation pattern on body tissue and voxel body, respectively. Figure 14 depicts that the antenna radiates outwards and mainly in a broadside direction. Compared with the antenna’s radiation pattern in the air, the radiation pattern on the tissue body has become more directive and in the broadside direction. In addition, the radiation pattern at 3.4 GHz is instead towards 180° than 5.6 GHz, which is altered by almost 20 degrees. The antenna lost many side lobes and back radiation after getting close to the body tissue.

Furthermore, it is because body tissue acts as a reflector that reflects the wave, not towards the body at a lower frequency. As a result, the antenna’s steering range has become narrower than the one in the air (the antenna’s radiation pattern on body tissue and voxel was investigated and examined to show the antenna’s capability to work at different frequencies and bands. It was also necessary to ensure that the antenna radiates primarily away from the body so as not to interfere with the existence of the body). Figure 15 illustrates the radiation pattern of the antenna on the body voxel. It should be mentioned that only ¾ of the body voxel is considered in the simulations due to accelerating the process and reducing the mesh cells in the CST studio during the assessment. Similar to the radiation pattern of the body tissue, the antenna’s radiation pattern on the voxel body is all outward. The radiation patterns are mostly towards 60° at lower bands on the arm, 120° on the shoulder, and 60° on the chest. Moreover, in all locations, the radiation patterns are outward of the body, and the radiation for higher and lower bands is not altered dramatically (3.4 GHz and 5.6 GHz).

Another factor that should be considered for evaluating the wearable antenna and communications is the SAR parameter. The SAR values should be less than 2 and 1 W/kg^3^ based on American and European standards, respectively [85]. Figure 16 and Figure 17 show the SAR variation of the proposed antenna on human tissue and voxel body (arm, chest, shoulder). They offer a maximum value of 1.44 on tissue and 2.6 on human voxel. The values did not exceed the standard values even when the antenna distance was between 5 and 8 mm. The exact values of the SAR factor are presented in Table 3. All SAR values are below standard levels and in the acceptable range [87]. Therefore, it can be concluded that the antenna cannot be affected by the presence of the human tissue and voxel body. It should be mentioned that the gaps between the antenna, the human tissue, and the voxel are almost 5 mm and 8 mm, respectively. 

The antenna’s simulated and measured radiation characteristics for on-body (on arm, chest, and shoulder) and off-body conditions are presented in Table 3 and Figure 18. The simulated and measured assessment of the antenna shows a decent agreement. In addition, the antenna’s radiation efficiency is greater than 81% for all BW under both off-body and on-body situations. The proposed antenna depicts a maximum gain of less than 8.

After examining the antenna’s SAR values and proving that it works well on the human body with the least SAR and effects, the antenna’s robustness towards bending is assessed by checking the S-parameter results. The button and LWA components of the antenna were bent toward the X- and Y-axes individually to determine whether the antenna performs correctly when each section bends or crumples after being worn to study the effect of real-world wearable conditions on the antenna performance. Figure 19 shows the robustness testing of the antenna in the direction of both X (Figure 19a,c)- and Y (Figure 19b,d)-axes. As illustrated in Figure 19, the button and LWA elements of the antenna are bent up to 20 and 60 degrees, respectively, in both the X- and Y-axes. 

Figure 20 shows the suggested antenna’s S-parameter findings when its button component or LWA part is bent. It demonstrates how the antenna’s reflection and transmission coefficients change slightly after turning. After bending, some stopbands were produced in the working BW Nevertheless, the band still contains both resonances crucial for the above applications, and the device continues to function effectively in those bands. Sn1: is the transmission results in S21 and S31 for different angles of a–d; Snn: is the reflection coefficient result from S11 to S33 for different angles of a–d shown in Figure 19. Figure 19a,b was bent to a maximum angle of 40 degrees, whereas Figure 19c,d was inclined to a maximum angle of 150 degrees. 

Three factors are investigated and analyzed in both on- and off-body circumstances to demonstrate the diversity of the antenna. Figure 21 depicts the diversity assessment of the MIMO antenna. The proposed MIMO antenna offers good diversity since the diversity gain (DG) is almost 10, and the envelope correlation coefficient (ECC) is less than 0.05. In addition, the AR is less than three at both resonances. It should be mentioned that all three parameters follow the same tendency for both conditions on and off-body. Furthermore, they were reduced when the antenna was close to the human tissue. Table 4 compares the proposed work with some recently similar three-port MIMO antennas designed for 5G and WBAN. 

## 4. Conclusions

A dual-band wearable three-port button MIMO antenna is proposed. With this antenna, a new dual-band MIMO simultaneous for WBAN and 5G communications was realized, where one three-spiral button antenna and LWA are utilized, yielding high diversity and higher channel capacity and data rate transmission. The two frequency bands of 3.25–3.65 GHz and 5.4–5.85 GHz required by WBANs, 5G, and sub-6 GHz are offered by the proposed antenna with a maximum of 94% of radiation efficiency and 7.2 dBi gain. A full ground attached to the LWA and the CSRR for the spiral antenna was implemented to obtain the wider BW, higher efficiency and gain, dual polarization, and miniaturized dimensions compared to several recent similar works. The antenna operates at sub-6 GHz and lower 5G communication bands. It functions well as a MIMO antenna with high diversity due to the isolation of more than −15 dB, an ECC less than 0.05, 10 dB of DG, AR < 3, and dual polarization. The proposed antenna’s high performances were evaluated after simulation by measurement for both off and on-body (on tissue and voxel body). The flexibility of the antenna was also examined by bending and tilting analysis. The excellent agreement between the simulated and measured results enables the antenna to be considered a liable candidate for WBAN applications and 5G communications. In addition, the proposed antenna is well suited for a MIMO system, which is confirmed by MIMO performance parameters such as ECC and DG. The proposed antenna can be a good candidate to ease communication among the soldiers or nurses in the field during the action, communications for laptop computers, air traffic, defense tracking, and the upper WLAN frequency range. 

## Figures and Tables

**Figure 1 sensors-23-01997-f001:**
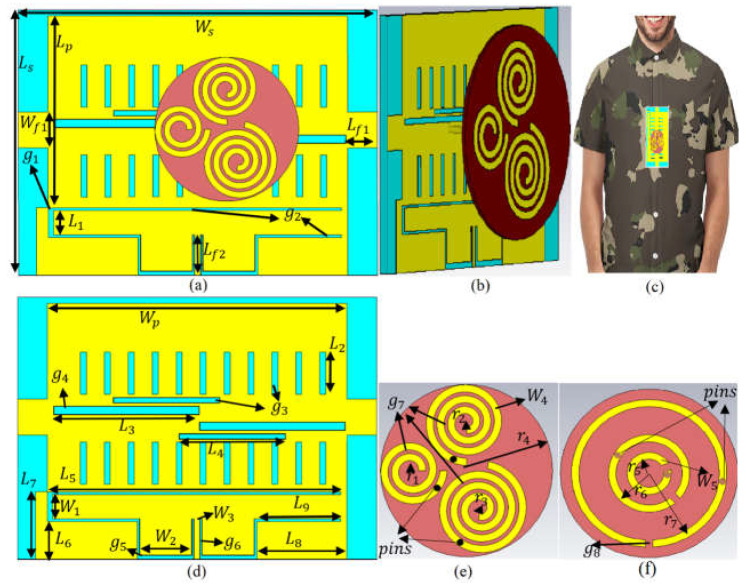
The proposed antenna’s simulated prototype ((**a**–**c**) perspective view; (**d**) wearable LWA; (**e**) spiral button antenna; (**f**) ground view of the button spiral).

**Figure 2 sensors-23-01997-f002:**
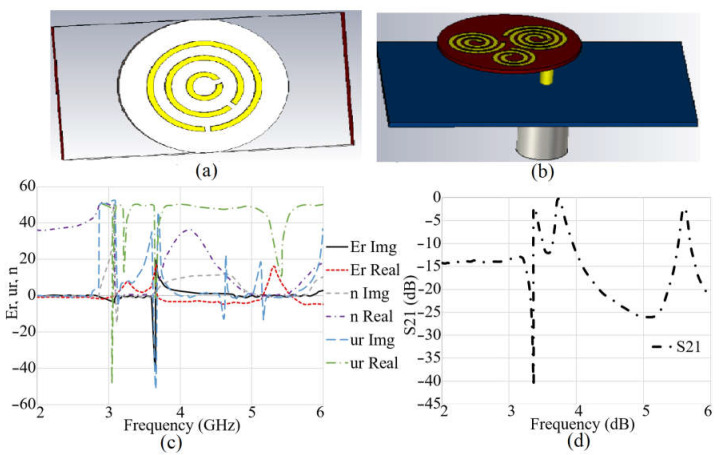
(**a**) SRR unit cell, (**b**) button antenna, (**c**) SRR unit cell S-parameters and extracted permittivity (er), permeability (ur), and refractive index (n) results, and (**d**) transmission coefficient (S21) result.

**Figure 3 sensors-23-01997-f003:**
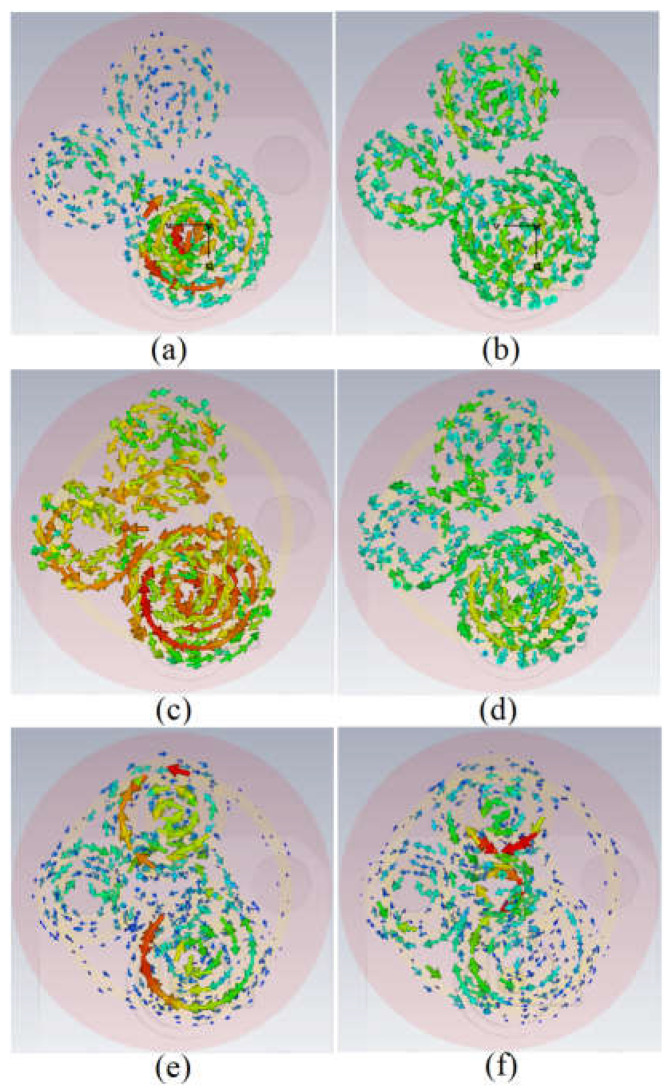
The surface current distribution of the button spiral antenna (**a**,**b**): no loadings at 3.4 GHz and 5.6 GHz; (**c**,**d**): after adding three rings at 3.4 GHz and 5.6 GHz; (**e**,**f**): after adding vias at 3.4 GHz and 5.6 GHz).

**Figure 4 sensors-23-01997-f004:**
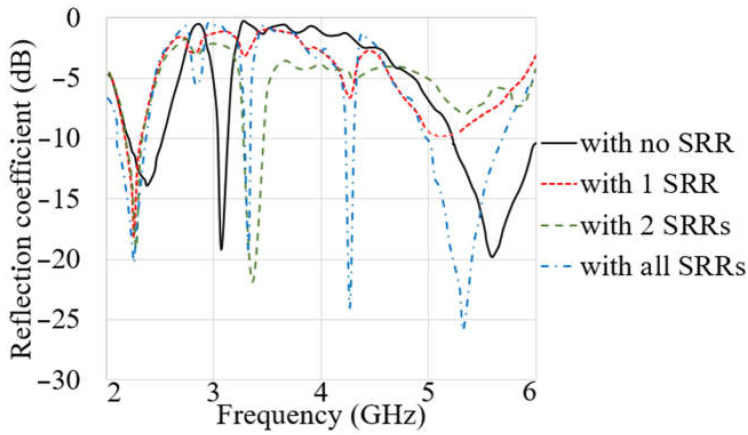
The reflection coefficient results with each of the SRRs and without SRR added to the button antenna.

**Figure 5 sensors-23-01997-f005:**
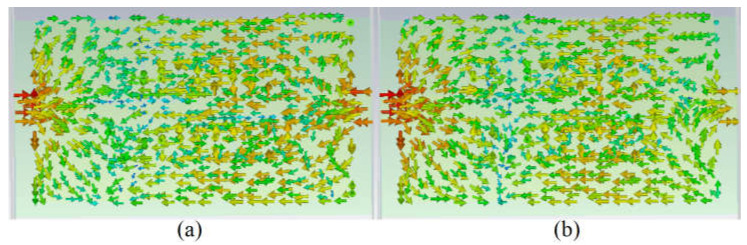
The surface current distribution of the LWA antenna ((**a**): at 3.4 GHz, (**b**): at 5.6 GHz).

**Figure 6 sensors-23-01997-f006:**
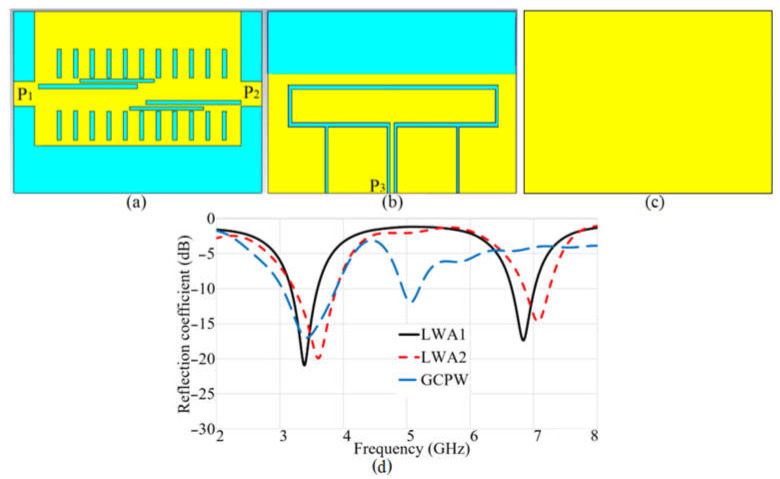
Each part of the MIMO antenna and its reflection coefficient results in (**a**) LWA 1 (P1) and LWA 2 (P2), (**b**) GCPW feed antenna, (**c**) ground view of the antenna, and (**d**) reflection coefficient results of each part of the MIMO.

**Figure 7 sensors-23-01997-f007:**
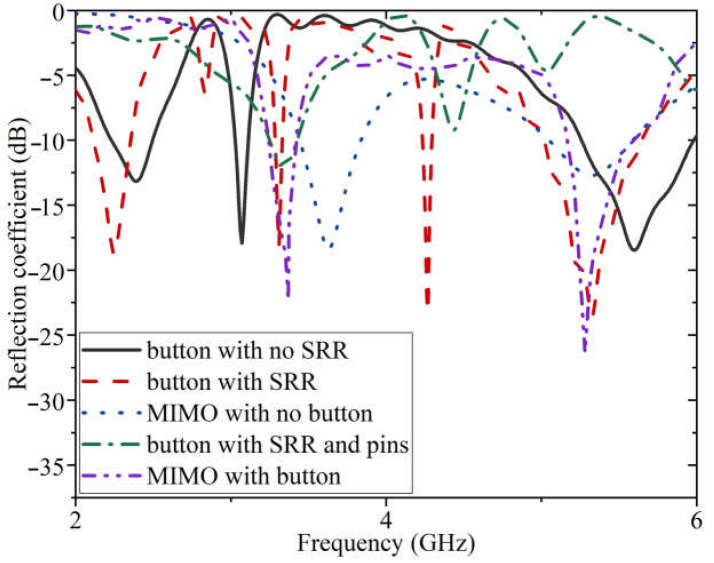
The reflection coefficient results of the antenna at each stage, before and after integration.

**Figure 8 sensors-23-01997-f008:**
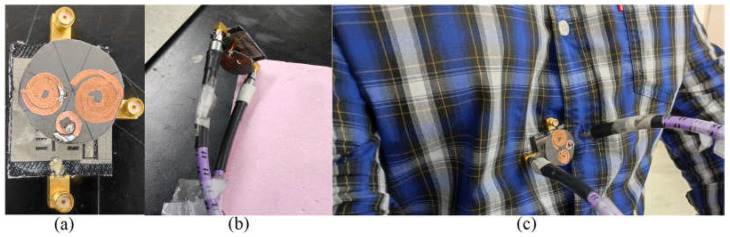
The measurement setup of the proposed antenna in both free space and on-body ((**a**) fabricated antenna; (**b**) measurement in free space; (**c**) measurement on chest).

**Figure 9 sensors-23-01997-f009:**
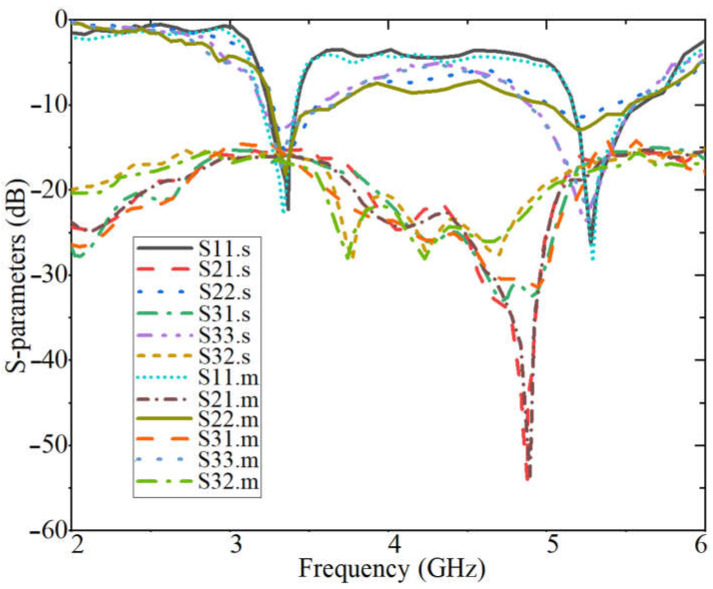
S-parameter results of the antenna in free space (off-body) (‘.s’ and ‘.m’ are the simulation and the measurement results, respectively).

**Figure 10 sensors-23-01997-f010:**
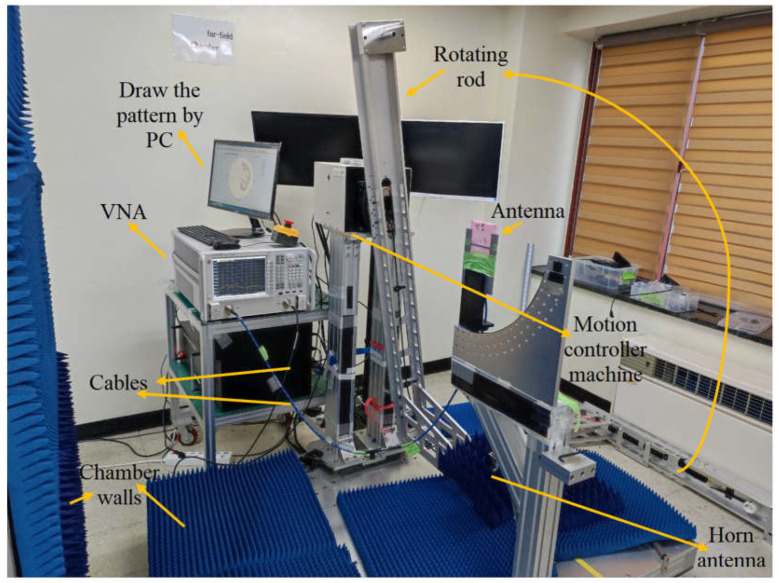
The measurement setup of the antenna to measure the radiation pattern.

**Figure 11 sensors-23-01997-f011:**
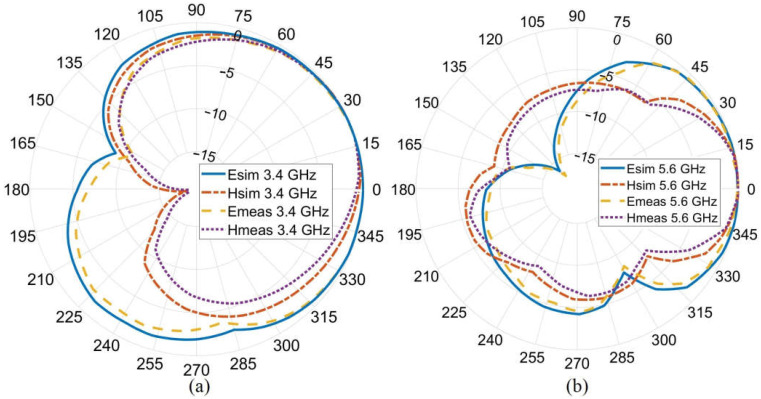
The simulated (sim) and measured (meas) radiation pattern (E-field (XZ plane)/H-field (YZ plane)) of the proposed antenna at (**a**) 3.4 GHz and (**b**) 5.6 GHz.

**Figure 12 sensors-23-01997-f012:**
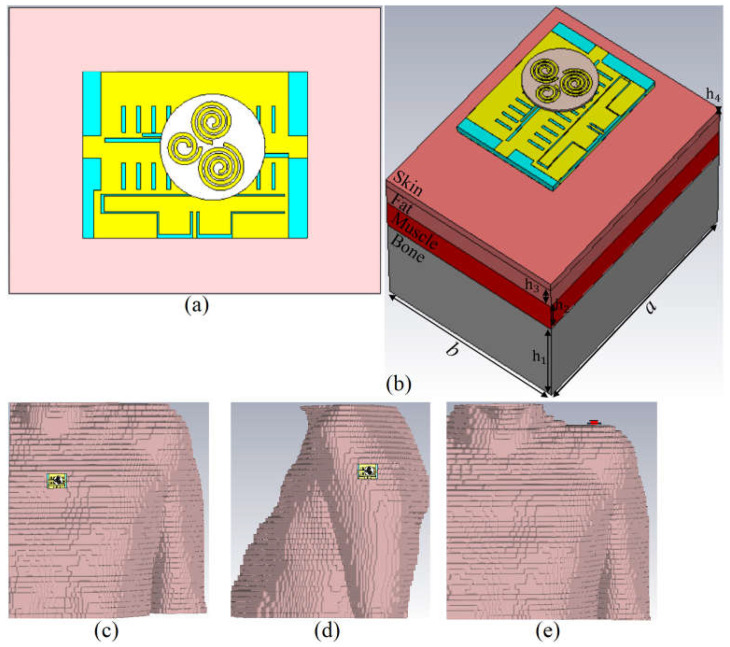
The simulation setup of the proposed antenna on body tissue cube (**a**,**b**) and voxel body (**c**–**e**).

**Figure 13 sensors-23-01997-f013:**
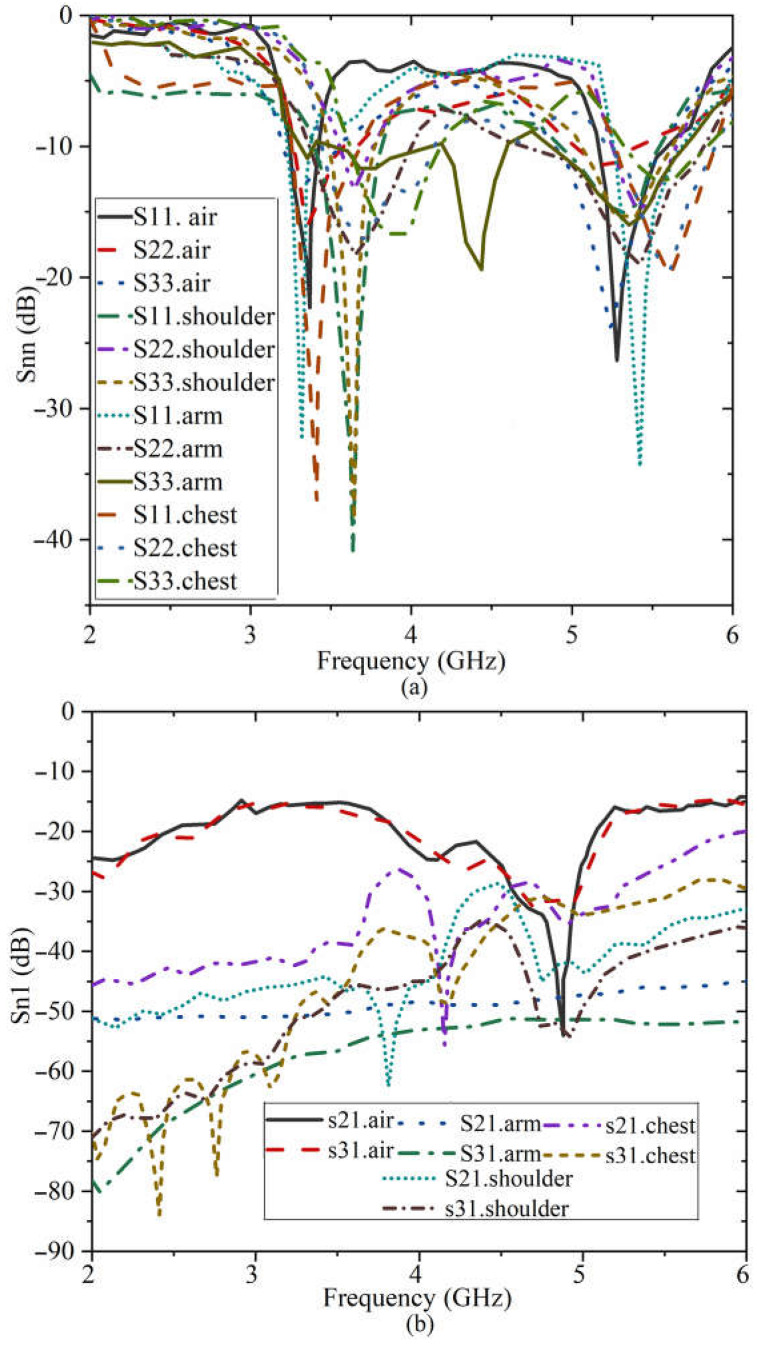
The S-parameter results of the proposed antenna on tissue body and voxel shown in Figure 12, (**a**): Snn is the reflection coefficient, and (**b**): Sn1 is the transmission coefficient.

**Figure 14 sensors-23-01997-f014:**
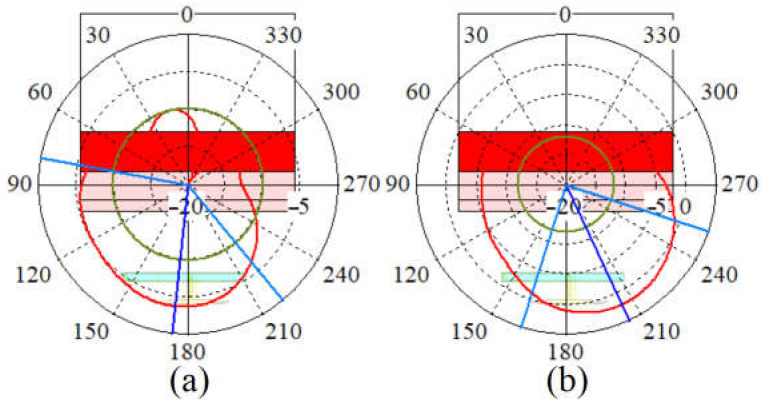
The simulated radiation pattern of the antenna on-body tissue at (**a**) 3.4 GHz, (**b**) 5.6 GHz.

**Figure 15 sensors-23-01997-f015:**
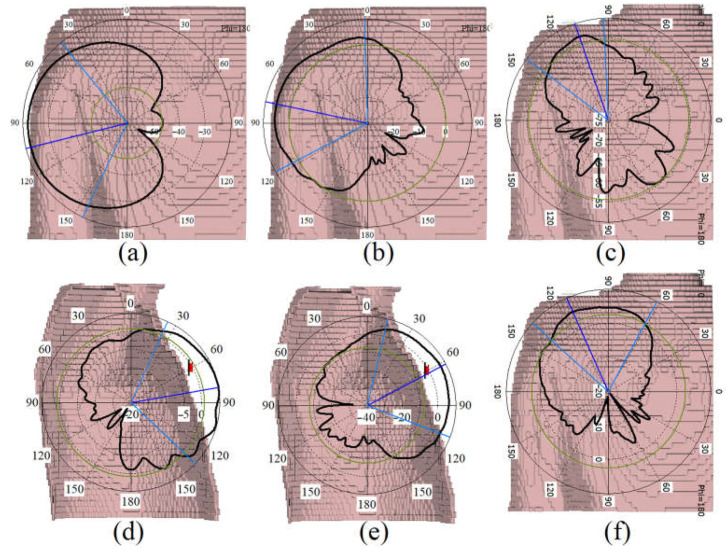
The simulated radiation pattern of the antenna on the arm (parts (**a**) at 3.4 GHz and (**b**) 5.6 GHz), on the shoulder (parts (**c**) at 3.4 GHz, (**f**) at 5.6 GHz), and on the chest (parts (**d**) at 3.4 GHz, (**e**) at 5.6 GHz).

**Figure 16 sensors-23-01997-f016:**
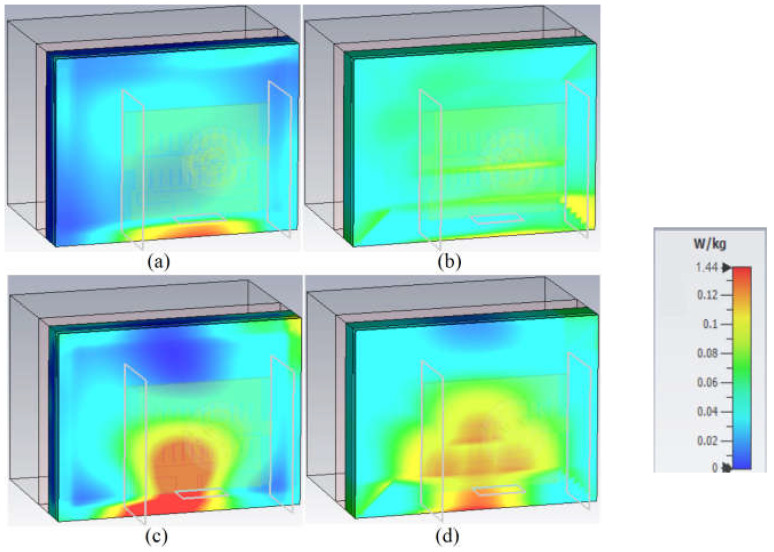
SAR evaluation of the proposed antenna at the proximity of human body tissue ((**a**) at 3.4 GHz, 1 g; (**b**) at 3.4 GHz, 1 g; (**c**) at 5.6 GHz, 10 g; (**d**) at 5.6 GHz, 10 g).

**Figure 17 sensors-23-01997-f017:**
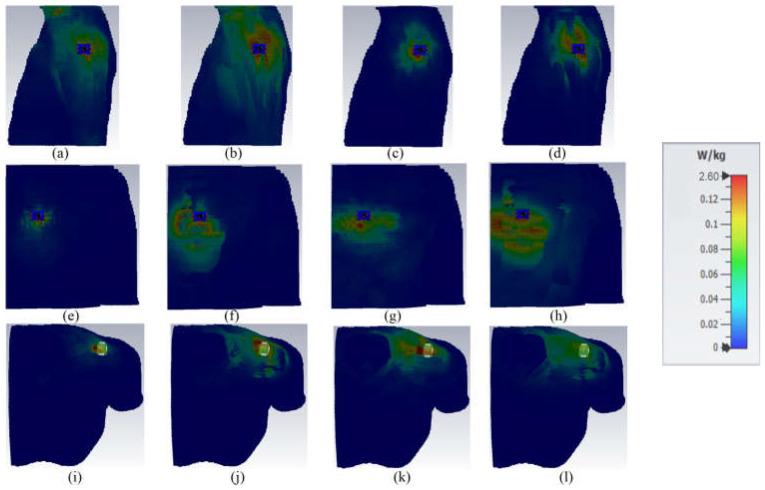
SAR evaluation of the proposed antenna at the proximity of voxel human arm, shoulder, and chest ((**a**,**e**,**i**) at 3.4 GHz, 1 g; (**b**,**f**,**j**) at 3.4 GHz, 10 g; (**c**,**g**,**k**) at 5.6 GHz, 1 g; (**d**,**h**,**l**) at 5.6 GHz, 10 g).

**Figure 18 sensors-23-01997-f018:**
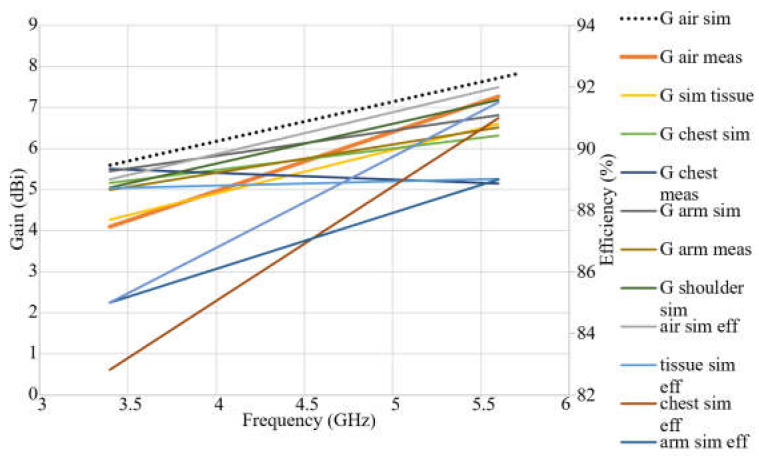
The simulated and measured gain and simulated radiation efficiency (‘sim’ is the simulation, ‘meas’ is the measurement, ‘G’ is the gain, and ‘eff’ is the radiation efficiency).

**Figure 19 sensors-23-01997-f019:**
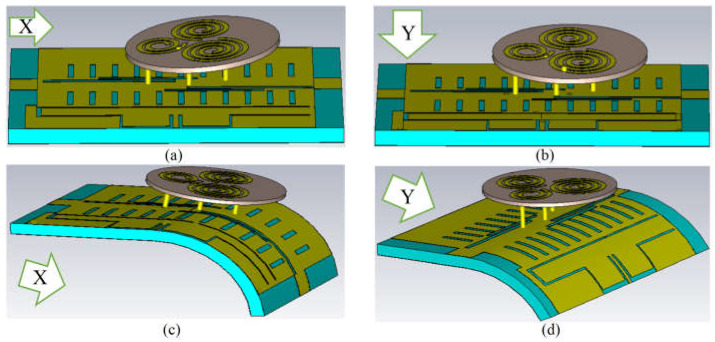
The crumpled antenna for both parts of the button and LWA towards the X (**a**,**c**)- and Y (**b**,**d**)-axis.

**Figure 20 sensors-23-01997-f020:**
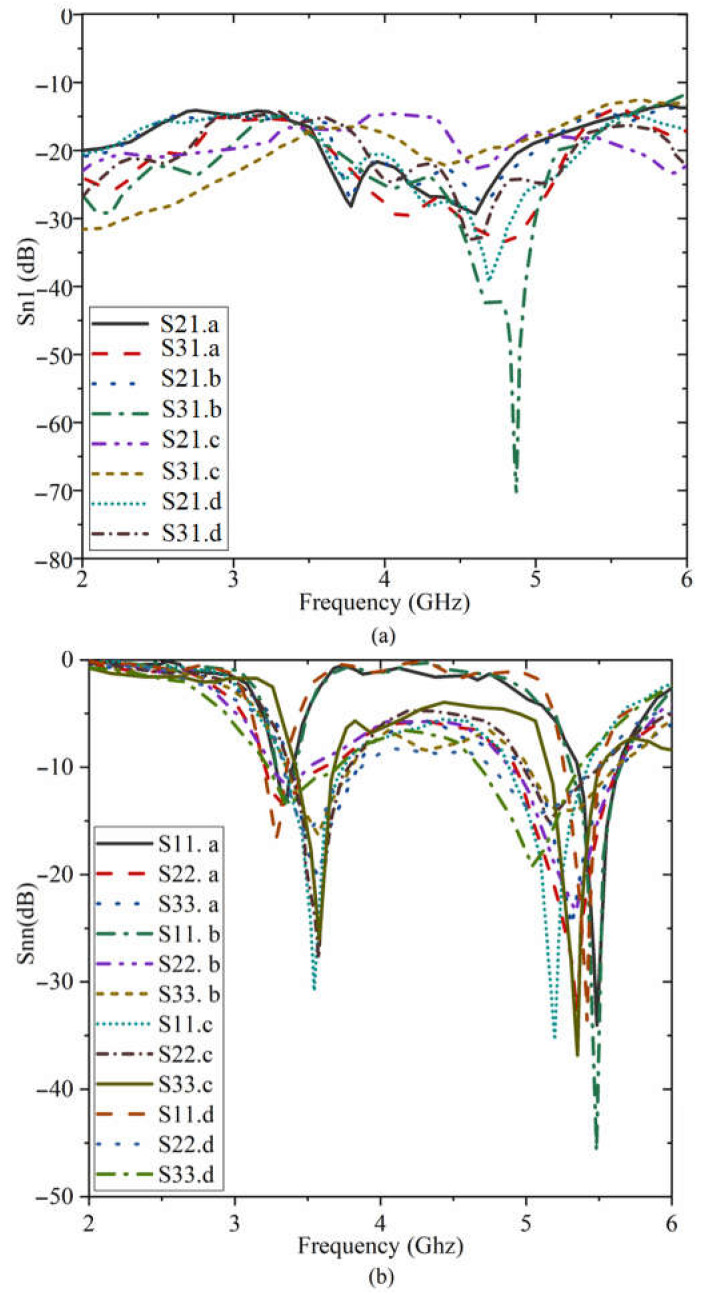
The robustness investigation of the proposed antenna in terms of S-parameters ((**a**) transmission coefficient results, (**b**) reflection coefficient results).

**Figure 21 sensors-23-01997-f021:**
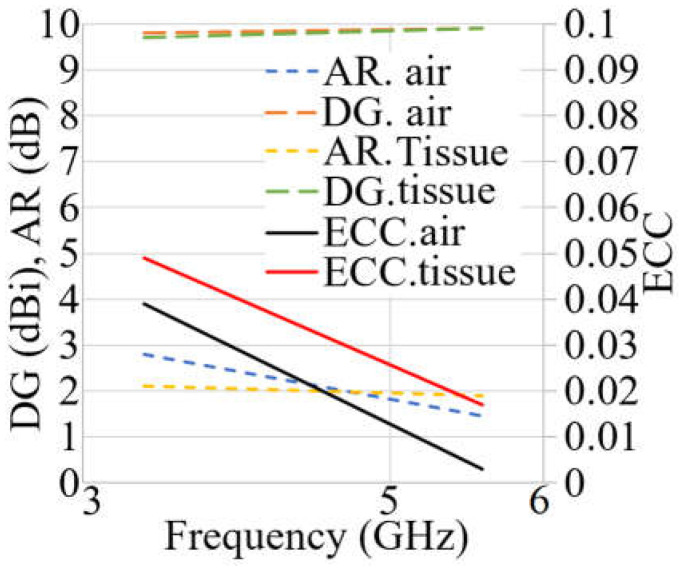
MIMO investigation results of the antenna in terms of DG, AR, and ECC.

**Table 1 sensors-23-01997-t001:** Comparison of flexible/wearable MIMO antenna in the literature.

	Param	fr (GHz)	Polarization	Peak Gain (dBi)	Size (mm) L × W	Designed Technique	Eff (%)	Material	Isolation (dB)
Ref									
[38]		2.45	CP	2.36	0.32λ_0_ × 0.32λ_0_	DGS	N/A	FR4	<25
[39]		5.8	N/A	<8.5	0.73λ_0_ × 1.1λ_0_	EBG, MTS	<83	Rogers 5880	<34.8
[40]		3.57	N/A	<5	0.70λ_0_ × 0.65λ_0_	Partial GND	N/A	Jeans	<19
[41]		2.45	N/A	0.5	0.22λ_0_ × 0.2λ_0_	Partial GND	<40	Rogers 5880	<30
[42]		6.5	N/A	5.88	0.7λ_0_ × 0.7λ_0_	Partial GND	N/A	Jeans	<22
[44]		6	Ver, Hor	4.62	0.52λ_0_ × 0.52λ_0_	Partial GND	93	Polyester	<17
[45]		1.1	N/A	7.5	0.19λ_0_ × 0.13λ_0_	Partial GND	<90	cotton and felt	<30
[46]		2.4	N/A	N/A	0.44λ_0_ × 0.13λ_0_	Stub at GND	<80	Jeans	<19
[47]		3.1	LP	3.41	1.1λ_0_ × 0.18.4λ_0_	Connected ground plane	89.3	Silicon rubber	<25
[48]		6	N/A	<8	0.76λ_0_ × 0.76λ_0_	CPW	<98	LCP	<22
[49]		3.55	Dual	<8.5	0.8λ_0_ × 0.8λ_0_	SIW, matching cavity	<93	Rogers RO4350	<18
[50]		5.3	CP	<4	0.87λ_0_ × 0.79λ_0_	Partial GND	N/A	FR4	<20

**Table 2 sensors-23-01997-t002:** The proposed antenna’s design parameters (mm).

Parameters	Final Values (mm)	Parameters	Final Values (mm)	Parameters	Final Values (mm)	Parameters	Final Values (mm)
$L_{s}$	22.50	$W_{s}$	33.00	$g_{1}$	0.50	$L_{f2}$	1.00
$L_{p}$	15.72	$W_{p}$	25.00	$g_{2}$	0.25	$r_{1}$	0.25
$L_{1}$	2.10	$W_{1}$	1.00	$g_{3}$	0.50	$r_{2}$	0.50
$L_{2}$	3.50	$W_{2}$	0.50	$g_{4}$	0.50	$r_{3}$	0.25
$L_{3}$	7.00	$W_{3}$	1.00	$g_{5}$	0.50	$r_{4}$	6.00
$L_{4}$	1.30	$W_{4}$	0.50	$W_{f1}$	3.00	$r_{5}$	1.15
$L_{5}$	3.80	$W_{5}$	0.525	$L_{f1}$	2.50	$r_{6}$	2.25
$L_{6}$	3.3	$L_{7}$	6.3	$L_{8}$	7.5	$r_{7}$	5.50
$L_{9}$	7	$g_{6}$	0.35	$g_{7}$	0.25	$g_{8}$	0.50

**Table 3 sensors-23-01997-t003:** Antenna’s performance comparison in three different mediums as free space, body tissue, and voxel body (arm, chest, shoulder).

f_r_ (GHz)		3.4	5.6
Free space	Gain (dBi) Sim/Meas	5.7/4.1	7.81/7.24
	Eff (%)	89	92
On tissue	Gain (dBi)	4.27	6.6
	Eff (%)	88.71	89.01
	SAR (1 g/10 g)	0.074/0.025	0.265/0.138
On chest	Gain (dBi) Sim/Meas	5.16/-	6.31/5.15
	Eff (%)	82.81	90.99
	SAR (1 g/10 g)	0.05/0.02	0.002/0.001
On arm	Gain (dBi) Sim/Meas	5.45/-	6.82/6.51
	Eff (%)	85	89
	SAR (1 g/10 g)	0.031/0.021	0.36/0.191
On shoulder	Gain (dBi) Sim/Meas	5.05/-	7.18/-
	Eff (%)	85	91.5
	SAR (1 g/10 g)	0.512/0.534	0.36/0.355

**Table 4 sensors-23-01997-t004:** Comparison of the proposed fabric antenna with previously reported work on MIMO antennas (three-port/wearables).

References	Freq. (GHz) Off/On	Dual Band	Polarization	Max Eff (%)	Max Gain (dBi)	Dim (mm) L × W	DG	Tilting/Bending Analysis	ECC
[1]	3.5	No	N/A	82	N/A	0.81λ_0_ × 0.4λ_0_	N/A	No	<0.17
[2]	2.55	No	Orthogonally	74	3.2	0.43λ_0_ × 0.43λ_0_	Yes	No	<0.36
[5]	2.4	No	triple	51.4	3.8	0.42λ_0_ × 0.42λ_0_	N/A	No	<0.37
[7]	2.3, 2.5, 2.4, 3.4–3.6	Yes	N/A	70	5	0.50λ_0_ × 0.65λ_0_	N/A	No	<0.1
[88]	5.8	No	LP/CP	85	4	0.56λ_0_ × 0.92λ_0_	N/A	No	<0.01
[89]	4.8–10.6, 8.1–10.8, 7.2–9.8	Yes	N/A	70	7	0.45λ_0_ × 0.45λ_0_	No	No	<0.08
[90]	2.74–12.33 GHz	No	N/A	N/A	6.9	0.50λ_0_ × 0.31λ_0_	9.9	No	<0.025
[91]	3.2–3.85, 5.15–5.35, 5.72–5.82	Yes	LP	93	N/A	0.26λ_0_ × 0.23λ_0_	10	No	<0.06
This work	3.25–3.65 GHz, 5.4–5.85 GHz	Yes	LP/CP	92.7 at 5.6 GHz	7.2 at 5.6 GHz	0.37λ_0_ × 0.25λ_0_	10	Yes	<0.05

## Data Availability

Not applicable.

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
