# Peer review of "A Miniaturized Full-Ground Dual-Band MIMO Spiral Button Wearable Antenna for 5G and Sub-6 GHz Communications"

_sensors, 2023, doi:10.3390/s23041997_

Round 1
Reviewer 1 Report
The authors propose MIMO antenna wearable antenna for 5G applications. Some comments are given as follows:
1. Why IoT has been mentioned in the abstract? It should be removed.
2. The SRR unit cell analysis is not clear? Extraction of real and imaginary parts is needed for both mu and epsilon. Moreover, the refractive index is not mentioned in the manuscript. However, the stop band (s21) should be considered for the resonance in unit cell analysis NOT s11, please clarify.
3. What is the benefit of using more than one unit cell? please justify using new fig. with one, two,.... and without SRR
4. Please rearrange your figures accordingly. Such as fig. 5 coming before fig. 4 in the manuscript.
5. Some sentences are well-known and NO need to be mentioned in the text such as
After attaching the antennas to the VNA's terminals using cables, it is necessary to calibrate the VNA using the calibration kits according to the diameter of the antenna's SMA.
6. E field (sim and measured) in Fig. 9 results are not agreed. Please remeasure and double-check. However, add the axis plane or clarify which is cross or polar co polarization.
7. Try to avoid the word novel in the conclusion.
Author Response
Thank you very much for your comments.
The paper was rechecked by an English writer and Grammarly premium.

Reviewer 2 Report
The main contributions and motivations for the present work have to be clearer.
In introduction the authors can come up with the existing survey works on the similar topic, probably summary table.
The research method is not clear, please clarify the research method involved.
Why the authors have chosen only this model why not others must be more clear.
There are many typos and grammatical errors try to proof read the paper.
Before using the annotations the authors should specify the and mention, kindly check it
Figure 6. The reflection coefficient results of the antenna at each stage, before and
after integration It depicts the reflection coefficient results of the button antenna with and without
SRR rings, the button antenna with SRR and pins, and the MIMO antenna with and without integration of the button antenna. The authors should clearly mention why the proposed model is better
Figure 14. SAR evaluation of the proposed antenna at the proximity of human body tissue (a: at 3.4 GHz, 1g; b: at 3.4 GHz, 1g; c: at 5.6 GHz, 10g; and d: at 5.6 GHz, 10g) needs more explanation
The authors can refer
Autonomous Vehicles in 5G and beyond: A Survey
Akram, Junaid, Awais Akram, Rutvij H. Jhaveri, Mamoun Alazab, and Haoran Chi. "BC-IoDT: blockchain-based framework for authentication in internet of drone things." In Proceedings of the 5th International ACM Mobicom Workshop on Drone Assisted Wireless Communications for 5G and Beyond, pp. 115-120. 2022.
Author Response

(The authors gave the same response as above.)

Reviewer 3 Report
In this paper the authors have designed and fabricated the small MIMO spiral antenna and leaky wave antenna with dual band response. The integration of the antennas was presented and their results tested practically. The presented work looks interesting. However, following suggestions need to be addressed.
1 The English writing of the article is not up to the scientific standards. It must be proof read by a native speaker.
2 In the abstract line 20 and 21 “The designed antenna with 3.25 to 3.65 GHz and 5. 4 to 5. 85 GHz operational bands wireless local area network (WLAN) frequency (5.1 - 5.5 GHz), the fifth generation (5G) communication band, and the internet of things (IoT).” WLAN work normally at 2.4 GHz, how you claimed WLAN? Do you think your designed antenna really covers 5G Spectrum and IoT applications? Pls think on it, revise it carefully and give me the proper justification. In that case, your title of the manuscript must be revised. Pls chose the suitable concise and clear title which supports the content given in the article.
3 The Introduction section must be revised. The contributions of the article should be highlighted in bullets.
4 There is need to add the related work section in which the prior state-of-the-art work concisely explained and briefly explain the advantages of the proposed work in comparison to the literature.
5 what is the reason of selecting particular type of antennas? Why the authors only select the spiral button and leaky wave antenna and integrate together? Isn’t that increase the design complexity and increase the fabrication cost?
6 There are lot of equations presented i.e Equation 1 – 11. Have you derived these equations? If yes then pls provide justification and proof.
7 Table 1, column 2, 4 and 6, write the “Final Values”, instead of “values (mm)”
8 Figure 5 a – b shows the designed leaky wave antennas. I didn’t see the ground plane on the other side of the substrate. Can you pls show the bottom view of the leaky wave antennas?
9 Figure 5c, why the GCPW results are not following the same trend as the results of LWA1 and LWA2?
10 pls revise Figure 8. Carefully check again the S parameters results Simulated and measured.
11 Figure 9 (a – b), why there is much difference in simulated and measured results? Author should comment on this.
12 Figure 11 (a – b)a and Figure 17 (a – b) in the horizontal axis, Snn (dB) and Sn1 (dB) why is like this? Isn’t that same as reflection coefficient?
13 Figure 12 (a – b) shows the radiation pattern performance of antenna on-body tissue at two resonances and Figure 13 (a – f) shows the performance of radiation patterns on voxel body. Why the pattern performances are different? What would be the reason to show the performance at two resonances in Figure 12 and six resonances in Figure 13?
14 Pls provide the photo of the antenna while measuring it inside the chamber room.
15 Further, explain how you measured the manufactured antenna. Provide and portray the complete visual microwave setup.
16 In table 2 and table 3, the gain and efficiency performance while taking different cases are given. Pls plot and show the simulated and measured results of the gain and efficiency.
17 Table 3, pls write the electrical dimensions in λ and calculate it at the lowest resonance of the proposed antenna. Revise the table and do it throughout the article.
18 What are the future perspectives of this study? Pls explain it in the conclusion section of the article.

Author Response

(The authors gave the same response as above.)

Round 2
Reviewer 1 Report
Thank you, all my comments have been addressed
Reviewer 3 Report
Thanks for incorporating my comments very seriously. However, pls carefully proof read the paper before publication. full form of WiMAX "Worldwide Interoperability for Microwave Access", pls write it correctly.